# Girls’ Empowerment and Adolescent Pregnancy: A Systematic Review

**DOI:** 10.3390/ijerph17051664

**Published:** 2020-03-04

**Authors:** Dumisani Enricho Nkhoma, Chia-Ping Lin, Hexin Latumer Katengeza, Charles Jenya Soko, Wanda Estinfort, Yao-Chin Wang, Shing-Hwa Juan, Wen-Shan Jian, Usman Iqbal

**Affiliations:** 1Master Program in Global Health and Development Department, College of Public Health, Taipei Medical University, Daan District, Taipei 106, Taiwan; m536107003@tmu.edu.tw (D.E.N.); m536107001@tmu.edu.tw (C.-P.L.); m536107004@tmu.edu.tw (H.L.K.); cjenyasoko@gmail.com (C.J.S.); wandaestinfort@gmail.com (W.E.);; 2Nkhata Bay District Hospital, Nkhata Bay District Health Office, P.O. Box 4, Mkondezi, Nkhata Bay, Malawi; 3Department of Emergency, Min-Sheng General Hospital, Taoyuan District, Taoyuan 330, Taiwan; d110099002@tmu.edu.tw; 4Graduate Institute of Biomedical Informatics, College of Medicine Science and Technology, Taipei Medical University, Daan District, Taipei 106, Taiwan; 5Graduate Institute of Injury Prevention and Control, College of Public Health, Taipei Medical University, Xinyi District, Taipei 106, Taiwan; 6International Center for Health Information Technology (ICHIT), Taipei Medical University, Daan District, Taipei 106, Taiwan; 7Department of Administration, Yuan’s General Hospital, Lingya District, Kaohsiung 802, Taiwan; 8School of Health Care Administration, College of Management, Taipei Medical University, Daan District, Taipei 106, Taiwan; jj@tmu.edu.tw; 9PhD Program in Global Health and Health Security, College of Public Health, Taipei Medical University, Daan District, Taipei 106, Taiwan

**Keywords:** girl, empowerment, economy, education, policy, community, adolescent, pregnancy, global health, systematic review

## Abstract

*Background*: 21 million girls get pregnant every year. Many initiatives are empowering girls. Various studies have looked at girl empowerment, however, there is contradicting evidence, and even less literature from developing countries. *Methods*: We searched articles published between January 2000 to January 2019. We followed Preferred Reporting Items for Systematic Reviews and Meta-Analyses (PRISMA) guidelines and registered our protocol on the International Prospective Register of Systematic Reviews PROSPERO (CRD42019117414). Nine articles were selected for review. Quality appraisal was done using separate tools for qualitative studies, cohort and cross-sectional studies and randomized control trials. *Results*: Eight studies included educational empowerment, four studies included community empowerment, three studies included economic empowerment, while two studies discussed policy empowerment. Three studies were of fair quality; two qualitative and one cross-sectional study were of high quality, while three studies had low quality. *Discussion*. Studies showed a favorable impact of girl empowerment on adolescent pregnancies and risky sexual behaviors. Education empowerment came through formal education or health systems such as in family planning clinics. Community empowerment was seen as crucial in girls’ development, from interactions with parents to cultural practices. Economic empowerment was direct like cash transfer programs or indirect through benefits of economic growth. Policies such as contraceptive availability or compulsory school helped reduce pregnancies.

## 1. Introduction

According to the World Health Organization (WHO), 21 million girls aged between 15 and 19 years and 2.5 million under 16 years give birth each year globally [1]. For adolescents, pregnancy is associated with a higher risk of complications such as puerperal endometritis and systemic infections compared to women aged 20–24 years. These complications are a major cause of death among girls within this age group [1].

Adolescents are young people aged between 10–19 years [2]. During this period, an individual transforms from childhood to adulthood and it is characterized by physical and psychological growth [3]. Due to the physiological and psychological changes that take place, adolescents are interested in exploring the world around them, in which some become sexually active. This puts them at a risk of sexually transmitted infection (STIs) and pregnancy, as some young people may lack often adequate knowledge of safe sex [4].

Usually, adolescent mothers have a tough time nurturing their children [5]. Later in life, these children have higher neonatal morbidity, difficulties reaching developmental milestones, and probably do not reach their full potential in life [6,7]. In addition, adolescent mothers, at times, are less likely to earn more than their peers [7]. The above situation usually leads to un-empowered women who are often at risk of, and unable to protect themselves from, abuse in relationships [8]. These effects tend to culminate into big losses economically for individuals and countries. Thus, governments and other institutions rollout strategies to empower adolescent girls and reduce negative reproductive or sexual health outcomes. It has also been reported that reducing teenage birth rate is associated with the rate of economic change, which may contribute to sustainable development [9].

Empowerment is a process of awareness and capacity building that leads to greater participation, decision-making power, and transformative action [10]. There are different types of empowerment that include educational, economic, policy, and community support [11]. In this review, we define empowerment for adolescent girls to include community, educational, economic, and policy support. Through empowerment, adolescents and women are equipped with knowledge and skills which enable them to make informed choices and take control of decisions that affect many aspects of their daily lives, including sexual and reproductive health. Thus, efforts that empower adolescents are crucial in reducing adverse sexual and reproductive outcomes such as adolescent pregnancies [12,13,14].

Sex education is one strategy that has been used to empower adolescent girls [11]. Sex education programs have been associated with better adolescent reproductive health outcomes and knowledge [15,16]. This type of empowerment equips adolescents with vital knowledge and improved self-concept that aide them in making crucial decisions about their reproductive life [17]. The rate of adolescent pregnancy varies according to economic status. Thus, economically empowered girls often have lower rates of pregnancy than those not economically empowered. Economic initiatives, however, have helped to reduce the rates of adolescent pregnancy in low-economic populations [11,18]. In general, economic empowerment affects adolescent pregnancy rates both directly and indirectly through, among other things, increased decision-making power structure in relationships, and access to contraceptives [19,20,21]. For example, a study in Uganda used vocational training and sex education. Results showed a 32% increase of girls’ participation in economic activities as well as a 26% decrease in the risk of adolescent pregnancy [18]. Apart from vocational training, other forms of economic empowerment interventions are microfinance and cash-transfers.

The above empowerment strategies target their effect at an individual level. However, nationally and later, global reduction in adolescent pregnancies would require empowerment strategies that take into account the community in which these girls are living in as well as government policies that guide them. In several communities, cultures that limit roles for adolescent girls and increase exposure to risky sexual behaviors have been noted [22,23]. Such challenges to adolescent health are more likely to be solved collectively at a community level. In addition, family support is crucial and may be key as to whether adolescent girls are truly empowered and progressive later in life and relationships [24]. While at a government level, policies such as youth-friendly services and the availability of contraceptives would create a youth-friendly environment in which any empowerment effect is more likely to yield positive results [22].

However, girl empowerment is not always associated with successful reduction of adolescent pregnancy rate [25]. Policies which are not well implemented could backfire. For example, despite rolling out adolescent-friendly services in health facilities, adolescents may be reluctant to access them if health workers are not sensitive [26,27]. On another note, under the Sustainable Development Goals (SDGs) target 3.7 and 5.3, renewed efforts are being put in place to empower adolescent girls and give them a brighter future. Therefore, it is important to gather evidence for empowerment strategies that specifically affect adolescent pregnancy, especially in Low and Middle Income Countries (LMICs) where the majority of adolescents dwell [28].

We therefore plan to carry out a systematic review of the literature on how girl empowerment influences pregnancy rates among girls between the ages of 10 and 19. In this study, forms of girl empowerment include economic, educational, community, and policy support, which is unique because most reviews focus only on specific form of girls’ empowerment, such as education.

## 2. Materials and Methods

### 2.1. Search Strategy

We conducted a literature search in PubMed, Web of Science and Scopus databases for studies published between January 2000 to January 2019, according to Preferred Reporting Items for Systematic Reviews and Meta-Analyses (PRISMA) guidelines [29] with the following keywords: ‘girl’, ‘women’, ’female’, ‘empower,’ ‘empowerment’, ‘economy’, ‘education’, ‘support’, ‘policy’, ‘community’, ‘teenage’, ‘adolescent’, ‘pregnant’, and ‘pregnancy’. The terms were searched in the following combination: TITLE-ABS-KEY (girl* OR women* OR female*) AND TITLE-ABS-KEY (empower* OR empowerment*) AND TITLE-ABS-KEY (economy* OR education* OR support* OR policy* OR community*) AND TITLE-ABS-KEY (teenage* OR adolescent*) AND TITLE-ABS-KEY (pregnant* OR pregnancy*). In PubMed, we used MeSH terms to search the key words and used Booleans of ‘AND’ and ‘OR’. In Web of Science, we used Topic (TS) to search the keywords using Booleans ‘AND’ and ‘OR’. In Scopus, we used the Booleans ‘AND’ and ‘OR’ to search the keywords using title (TITLE), abstract (ABS), and keyword (KEY) (TITLE-ABS-KEY). We registered our protocol on PROSPERO (registration number CRD42019117414).

### 2.2. Eligibility Criteria

The inclusion criteria were (1) peer-reviewed, (2) full-text articles, (3) those with a clear definition of female adolescent according to the WHO definition of adolescent (age 10–19 years), and (4) studies published between January 2000 and January 2019. In this review, articles were excluded for the following reasons: (1) they were duplicates, (2) the articles were a review paper, an editorial and/or a commentary, (3) the article was a non-empirical study, (4) the studies did not report an association between girl empowerment through community, educational, policy, or economic support and adolescent pregnancy, and (5) studies were not published in English.

### 2.3. Selected Studies

The review includes 18 years of research, including the most up to date literature. In total, 191 articles were published before the year 2000 and therefore were excluded. C.P.L, C.J.S, D.E.N, E.W, and H.L.K identified 123 records in Scopus, 16 records in PubMed, and 52 records in Web of Science. Sixty articles were excluded as follows; 31 articles were duplicates, 17 were review papers, editorials or commentaries, ten were study protocols, and two were not published in English, leaving 131 studies to be screened for eligibility.

In addition, 100 studies were excluded during screening titles and abstracts. Nineteen studies that did not clearly exhibit association between girl empowerment and adolescent pregnancy were also excluded, whereby girl empowerment came in the form of economic, educational, community, and policy support. This left 11 full-text articles from, which only nine articles were included in the review because two studies had with the female subsets who were already pregnant. This information is illustrated in Figure 1 below, following the PRISMA flowchart.

### 2.4. Summary Information

Study data were summarized into two categories, description of study characteristics and themes across studies. Studies were described according to the type of study design, geographical origin, and socio-economic setting. The empowerment of girls was summarized as community, policy, educational and economic, or combination thereof. Themes were constructed according to study findings and graded from most frequent to least occurring.

### 2.5. Quality of Studies

C.-P.L and D.E.N assessed the quality of studies. Cohort and cross-sectional studies were assessed using National Institutes of Health (NIH) Quality Assessment tool for Observational Cohort and Cross-Sectional Studies [30]. A set of 14 questions were posed with answering choices of ‘yes’, ‘no’ and ‘others’, whereby others stood for ‘not reported’, ‘could not determine’, or ‘not applicable’. Overall grade was assigned as good, fair, or poor according to assessment tool guidelines [30]. Qualitative studies were assessed using the Critical Appraisal Skills Program (CASP) tool [31]. This tool consists of ten guiding questions that help appraise studies, though originally not constructed to come up with scores, the tools’ questions helped reviewers to assign overall grade of high, low or fair quality. Risk of bias 2.0 tool (Cochrane Collaboration, London, UK, 2019) was used to assess bias for one randomized clinical trial (RCT) [32]. Detailed information on quality assessment is available in Appendix A.

### 2.6. Ethical Approval

Ethical approval will not be required because in this study we retrieved and synthesize the data from already published studies.

## 3. Results

Nines studies reporting an association between empowerment (education, policy, economic, social, and community) and adolescent pregnancy were identified. The United States of America (USA) and Kenya had two studies each [33,34,35,36] and the rest were conducted in Brazil, Canada, Nigeria, Nepal, and Sweden [37,38,39,40,41]. The distribution is relevant, as five of the nine studies were conducted in LMICs [35,36,37,39,40]. Of the nine articles, four were qualitative studies [33,35,38,41]. The first study, conducted in the USA, was part of a mixed method project, done to bring more explanation on quantitative findings. It involved school-going adolescent girls and measured the impact of involvement in sports on their sexual health [33]. The second qualitative study was conducted in the Northwestern territories of Canada. They identified themes (sexual health knowledge, relationships with the self and others, access to quality sexual health resources, and alcohol use/abuse) as perceived barriers and facilitators to empowered and safe sexual health [38]. It was done through face to face interviews, in order to comprehend barriers and facilitators of adolescent sexual health and behaviors. Another study involved focused group discussions of urban adolescent girls in Sweden [41]. The discussion revolved around contraception use, and facilitators of risky sexual behaviors. The last qualitative study was conducted in Nyanza province, Kenya [35]. Focus group discussions were conducted in low income communities involving adolescents, their guardians and other prominent community members. The study aimed at comprehending risky sexual behaviors of orphaned children living in poverty. It was discussed that poverty seemed to increase vulnerability to transactional sex, early marriage, sexual experimentation, increasing the risk of unintended pregnancies, and sexually transmitted infections (STIs), including human immunodeficiency virus (HIV).

Two studies used repeated cross-sectional data [37,39]. One study involved a nationwide representative sample of Brazil, looking for association of socio-economic factors and trends of adolescent pregnancies [37]. This study found a drop in live births in adolescent mothers between 2000 to 2011. The prevalence of live births to adolescent mothers was inversely proportional to socio-economic status measured by Human Development Index (HDI). The other cross-sectional study was conducted in Nigeria, measuring timing of first child births amongst women of childbearing age [39]. The median survival time to the first childbirth was three times higher in women with no formal education than in those with higher education. One study was a randomized control trial (RCT) of adolescents with Type 1 Diabetes [34]. It sought to compare preconception counseling to usual care involving scheduled clinic visits with standard counseling on the attainment of knowledge in order to help girls make informed reproductive health decisions. The intervention group had significant and sustained group and time interaction for benefits and knowledge of preconception counseling and reproductive health. Two were cohort studies [36,40]. A study in Nepal aimed at assessing the association between socio-economic factors pregnancy comparing adolescent mothers and mothers who had first birth after the age of 20 years [40]. Information was retrospectively collected on a nationwide representative sample, on which analysis showed adolescent mothers had comparatively low social class, literacy rate, contraception use, and knowledge before the first pregnancy. The other cohort study involved prospective observation of low-income adolescent girls equipped with self-defense strategies in order to protect themselves from sexual assault and harassment in a Kenyan area with high crime rate [36]. Results showed a higher decrease in sexual assault rates in the intervention group as compared to those in the control group. More details are provided in Table 1 below.

Eight (88.88%) articles in the review included educational empowerment, three (33.33%) articles included economic empowerment, two (22.22%) studies included community empowerment, two (22.22%) studies discussed policy empowerment, and two (22.22%) articles discussed community empowerment. Of the nine articles in the review, three concerned only one type of empowerment (education empowerment). Two articles involved a combination of education and community empowerment, one article had a combination of education and community empowerment, one had a combination of social, education and economic empowerment, one article had a combination of economic and policy empowerment and one had a combination of social, economic, and education. Education appeared to be the most prevalent and important type of empowerment, followed by community, economic, and policy empowerment, respectively.

### Quality Assessment

During quality assessment, three studies each were seen as high, low, and fair quality. High quality studies included two qualitative and one cross-sectional (repeated) [35,38,39]. A retrospective, one cross-sectional and one qualitative were judged as fair quality studies [37,40,41], while one mixed method, a prospective and a randomized control trial (RCT) were judged as poor quality studies [33,34,36]. All cross-sectional and cohort studies did not report pre-calculated sample estimations. Other limitations were recruitment of subjects from different populations or period [36,40] and failure to ascertain exposure measurement before outcomes [37]. For qualitative studies, common limitations on quality were not reporting an appropriate qualitative method [33], recruitment strategy, ascertaining the relationship between researcher and participant [33,41] and not using proper data collection in a way to address the study objective [33,35]. The included RCT was at a high risk of bias due to missing data, as well as selection of reported results [34].

## 4. Discussion

### 4.1. Summary of Evidence

In this review, evidence indicated that empowering adolescent girls may have a favorable effect in reducing adolescent pregnancy. Studies showed empowerment to be influential in changing adolescent sexual behavior, which later reduced rates of adolescent pregnancy. In qualitative studies, common themes supported the notion that girl empowerment would reduce adolescent pregnancy. Quantitative studies showed a favorable association between several empowerment strategies and reduction in risky sexual behaviors or outcomes, including adolescent pregnancies. Education empowerment was commonly implemented in most studies. In addition, most studies used a combination of empowerment types in order to achieve desirable outcomes. However, the evidence presented in this review involves nine studies, of which only three are of high quality and the rest are fair (moderate) and low.

### 4.2. Educational Empowerment

Educational empowerment has been described in two main ways. The first form of empowerment comes with the involvement of adolescent girls with conventional structures of education such as classrooms. In Nepal, compulsory education was seen as an effective option to postpone marriage and thus delay pregnancy [40], while in Nigeria, education was positively correlated to delaying the first childbirth [39]. Keeping adolescents in school has been linked with childbirth later in life, in turn having a desirable impact on their lives. Previous studies with different populations support this view. Observational studies have shown several correlations of higher education attainment, noted as educational empowerment with lower chances of adolescence pregnancy [42,43,44]. Conversely, adolescent mothers are less likely to complete formal education [43,45]. Though improving access to traditional education systems might seem crucial, other details such as quality need to be taken into account. Addition of sexual education classes to curricula may equip adolescents with the knowledge and thus empower them to make the right decisions regarding their sexual health. A study involving Canadian adolescents, in areas with the highest rates of teen pregnancies, reported that improving the content and delivery of sexual education might act as a facilitator to a positive, empowered and safer sexual health [38]. However, these findings should be interpreted in the context of several factors. In this study, accessibility and adequacy of such resources was another factor that was potentially affecting of adolescent sexual behavior. Quality of education could also be enhanced by the frequency of exposure to sexual health information. In a focus group discussion with urban teenagers in Sweden, researchers found that sex education sessions and gender related messages as a recurrent activity in the school curricula may help to empower young women and promote avoidance of risk-taking during sexual activity [41]. Sex education programs have been further noted in previous studies as fundamental in reducing adolescent pregnancy [46,47,48,49].

Reproductive health information can be disseminated not only in the school environment but also through sessions in clinical or community settings [50]. Of note, one study used this method to reduce sexual assault rates in Kenyan informal urban settlements. This in turn was associated with reduced adolescent reproductive outcomes and improved sexual assault disclosure [36]. Policies implementing adolescent friendly or reproductive health services may use this strategy to increase awareness or empowerment among participants.

### 4.3. Community Empowerment

Community structures and norms have been known to be associated with healthy adolescent sexual practices and behavior [51]. In a Brazil survey study, cultural and social factors were among key determinants associated with teen pregnancy [37]. Cultural practices could expose adolescents to early marriage, pregnancies and other adverse health outcomes [52,53]. Previous studies have highlighted problems adolescents face which may stem from the community such as pressure of early marriage and household chores [54,55]. That in mind, person to person interactions with those near to adolescent girls matter a lot. One of the most important interactions is between parent and child [56]. In a qualitative study, adolescents pointed out that the relationship between self and parents would be a vital determinant in reproductive health outcomes [38]. These suggestions are backed by a studies in which parenting styles were seen to affect adolescent health and academic outcomes [43,57]. In general, the community may have the potential to both increase the risk of adolescent pregnancy, at the same time, it might offer a protective environment that might spar youths to unprecedented growths in all aspects of life. This highlights the need to take into consideration social and cultural context before implementing girl empowerment strategies.

### 4.4. Economic Empowerment

Low economic status has been associated with unfavorable health outcomes [58,59]. A retrospective cohort study in Nepal found that low socio-economic status and poverty were associated with first childbirth during adolescence [40]. Another qualitative study collected views from participants who believed poverty both directly and indirectly influenced risky sexual behaviors among adolescents [35]. Their suggestions were consistent with a study in Tanzania that found transactional sex with older men was one of the few available sources of income that allowed adolescent girls to meet their basic needs, making this a common choice for many girls, even though it increased the risk of unintended pregnancy [60]. In Brazil, researchers found that high human development Index (HDI) ranking of communities was associated with low birth rates amongst adolescent teen girls [37]. Human development index is a geometric measure which is a product of health status; measured by the life expectancy at birth of a population, knowledge; measured by the average number of years a child spends in school and standard of living; measured by the gross national income per capita [61]. In spite of the HDI components, the study used this metric to assess different regions of the vast country of Brazil. Low economic status pushes adolescents to indulge in income generating activities which would at times be associated with risky sexual behaviors. Thus improving economic status through economic empowerment may have the potential to reduce adolescent pregnancy. In Malawi, for example, a conditional cash transfer program showed a link to better academic outcomes, better health, and decreased pregnancy rates and early marriage [62].

### 4.5. Policy Empowerment

Principles of action guiding governments can have a fundamental impact on its people. In terms of adolescent health, the main themes that stand out in this review are; education, contraceptive use, and their accessibility [38,39,40,41]. In the education system, regulations that ensure that girls are kept in school such as compulsory education [40], incorporating tailored sexual health education [38], and a supportive environment are likely to empower adolescents and positively affect their health. Contraceptive use and accessibility has always been seen as a cost-effective way to tackle adolescent pregnancy [63,64]. WHO advocates for increment in use of contraception as one of the six evidence based guidelines to reduce adolescent pregnancy. Alongside use of contraception, WHO advocates for policies to tackle coerced sex [64]. In a qualitative study, all participants admitted to knowing at least one person who was sexually abused. It was also reported that partners were more reluctant to use condoms or negotiate with them about protected sex [41]. This revelation reveals another problem. The problem is, despite availability of contraceptives, girls’ negotiating power in intimate relationships may be often reduced [65]. This issue needs to be taken into account when implementing any contraception program. In addition, even in the presence of strong policies, adolescent pregnancy rates have not reduced in many countries, especially in Lower and Middle Income Countries (LMICs), even though the majority of countries have protective laws stating such as legal marriage age [50,66]. Marriage or union is an important factor in tackling adolescent pregnancies. This is so since over 95% of adolescent pregnancies were noted to happen within union [67]. As such, community support, and cultural considerations through leaders and parents would be vital to compliment state-led policies. This, in conjunction with educational policies and contraceptive availability, would help prevent not only unintended pregnancies, but also intended pregnancies. For governments, high adolescent pregnancy rate and its consequences could deter development, while the right policies could save lots of money in health system costs if well implemented [63].

### 4.6. Lower and Middle Income Countries

Studies done in developing countries have stressed educational attainment and economic empowerment. Quantitative studies suggest that increased levels of education would be associated with lower levels of adolescent pregnancies, while economic empowerment would likely reduce risky sexual behaviors as well as pregnancies. Thus, a suggestion would be to consider educational and economic empowerment interventions in any programs targeting reduction in adolescent pregnancies in LMICs.

### 4.7. Limitations and Future Considerations

The review gets most of the evidence from non-experimental studies. Only one study was a randomized control trial, and one prospective study was included. A lack of such studies could be hypothesized as a result of ethical limitations in assessing the effect of empowerment on adolescent pregnancy. The small sample size was the biggest limitation, with even fewer studies reporting quantitative analysis. With this view in mind, the majority of the evidence discussed comes from studies with at least fair quality. However, the small sample size reduces the strength of evidence gathered. The number of studies might reflect the scope of the search strategy as only three databases were used or might indicate the little progress made in scientific literature in recent years regarding the topic. Future studies need to take into account inclusion of extensive grey literature which reports empowerment programs. Additionally, we could not explain synergy between two or more empowerment strategies or programs, a consistent finding in most included studies. However, the study managed to aggregate almost an equal number of studies from both high- and low-income countries, of which the latter has been scarce in literature. Future studies need to establish the generalizability of these findings to various populations. Other limitations relate to technicalities of the study. The review included studies from the year 2000–2018, full articles, only those in English and with a clearly stated association of empowerment and adolescent pregnancy. Future research may consider different dimensions of time and language.

### 4.8. Implications

This review offers insight into how we can tackle the daunting task to reducing adolescent pregnancies. Almost half of the articles included have offered qualitative evidence from the adolescent perspective. In addition, the evidence from this review is globally representative, combining studies from all continents and different cultures. Thus, the review offers readers and interested institutions with knowledge that may prove useful in designing interventions and programs. Reducing adolescent pregnancies is at the heart of sustainable development goals (SDG) 3 [66], which will guide many health professionals for more than a decade to come. The information aggregated in this review, strategies, or interventions could be built that support adolescent girls. Although combined strategies have not been thoroughly discussed in this review, we would suggest the use of combination of empowerment strategies. The different types of empowerment could be implemented simultaneously at different levels of the society. The aggregation of evidence-based points could be classified into individual, person to person, and community, as well a larger convention of societies, such as the state.

## 5. Conclusions

Adolescent pregnancy is a significant problem in the world. A lot of effort is being put in to reduce adolescent pregnancies through empowering the youths through education, the community, economic, and policy interventions. Educational empowerment is commonly applied. However, a combination of empowerment initiatives would seem beneficial, and further research should be conducted to clarify their role in reducing adolescent pregnancy.

## Figures and Tables

**Figure 1 ijerph-17-01664-f001:**
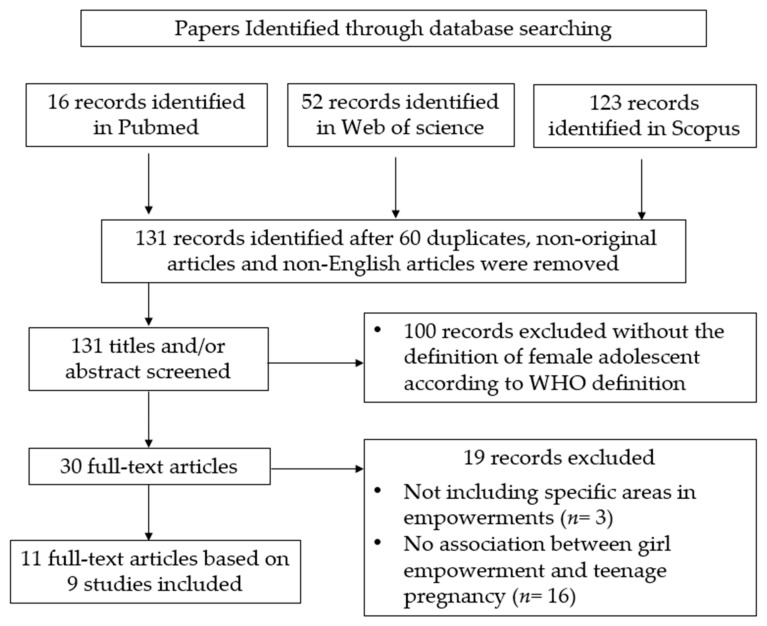
Process of identifying, collecting, and screening of articles according to PRISMA guidelines.

**Table 1 ijerph-17-01664-t001:** Characteristics of included studies.

Title	Study Authors /Reference ID	Year	Country	Study Design/Quality ^1^	Sample	Type of Empowerment	Empowerment (Measurement)	Major Findings
Adolescent women’s sports involvement and sexual behavior/health: A process-level investigation	Lehman et al. [33]	2004	United States	Mixed Methods investigation **Low**	176 adolescent women, 18–19 years	Educational and community empowerment	Self-empowerment/efficacy were asked to complete 4 scales (Masculinity, Femininity, Self-Reliance, Self-Efficacy)	Adolescent women’s involvement in organized team sports was significantly associated with the following: (a) sexual-risk-taking behavior (r = −0.34, *p* < 0.001), (b) sexual health-seeking behavior (r = 0.24, *p* < 0.01) ^2^, and (c) sexual/reproductive health (r = 0.21, *p* < 0.01).
Impact of a preconception counseling program for teens with type 1 diabetes (READY-Girls) on patient-provider interaction, resource utilization, and cost	Rodgers Fischl et al. [34]	2010	United States	Randomized controlled **Low**	88 teens with type I diabetes, (control *n* = 43), (intervention *n* = 45), age range 13.2–19.7, M age = 16.7	Economy, policy, educational empowerment	(a) Knowledge, attitudes, intentions and behaviors ^3^ (b) economic analysis (resource utilization, direct medical costs, break-even cost analysis	IG had significant and sustained group and time interaction for benefits and knowledge (preconception counseling and reproductive health. (3 months, *p* < 0.001, 9 months, *p* < 0.01 (benefits) *p* <0.001 (knowledge) and (intention). Direct medical costs were low.
Understanding orphan and non-orphan adolescents’ sexual risks in the context of poverty: A qualitative study in Nyanza Province, Kenya	Juma et al. [35]	2013	Kenya	Qualitative **High**	78 (53%) participants were adolescent aged 14–17 years (with 41 female (53%) and 37 male (47%)). 69 (47%) were caregivers	Economic and policy empowerment	This study used focus group discussions (FGDs) and key informant interviews (KIIs) with the following themes relating to poverty and risky sexual behavior.	Poverty seen as key predisposing factor to risky sexual behaviors. Poverty linked with factors that increase vulnerability to transactional sex, early marriage, sexual experimentation, and increased risk of unintended pregnancies and STI/HIV ^6^
Rape prevention through empowerment of adolescent girls	Sarnquist et al. [36]	2014	Kenya	A prospective cohort **Low**	1978 adolescents (intervention group)(SOC) ^4^ group = 428	Educational empowerment	The intervention was grounded in social learning theory and the health belief model and was adapted from existing empowerment and self-defense modules. (self-reported anonymous survey conducted)	Decrease in annual sexual assault rates (RR = 1.61 95% CI (1.26, 1.86) compared to SOC’s 1.02 95%CI (0.67, 1.57) ^5^. 52.3% from intervention group reported using learned skills to stop an assault.
Trends of adolescent pregnancy in Brazil, 2000–2011	Vaz et al. [37]	2016	Brazil	The cross-sectional (repeated) 2000-2011 **Fair**	The number of live births to women aged 10–19 years.	Social, educational and economic empowerment	Descriptive study to evaluate frequency of adolescent pregnancy correlating with human development index (HDI), conducted with data from the Brazilian Live Births Information System (Sinasc) of the Unified Health System (Datasus).	Drop in live births from adolescent mothers 23.5% (2000) to 19.2% (2011). HDI score were inversely proportional to the proportion of live births from adolescent mothers
Coming of age: how young women in the Northwest Territories understand the barriers and facilitators to positive, empowered, and safer sexual health	Lys et al. [38]	2012	Canada	Qualitative (semi-structured, face-to-face interviews) **High**	12 females aged 15–19	Community and educational empowerment	(a) self-perceived barriers facilitators to empowered, safe sexual health	4 themes influencing adolescents discussed: sexual health knowledge, relationships with the self and others, access to quality sexual health resources, and alcohol use/abuse.
Survival analysis and prognostic factors of timing of first childbirth among women in Nigeria	Fagbamigbe et al. [39]	2016	Nigeria	The cross-sectional (repeated) **High**	38,948 women aged 15–49 years were identified as eligible for individual interviews	Educational empowerment	The dependent variable in this study was age at first child birth while region and geographical zones of residence, education, religion, residence and ethnicity were the independent variables.	Median survival time to first birth (years): 27 (higher education) 18 (no formal education), adjusted hazard ratio (3.36 95%CI (3.17, 3.55) among women with no formal education compared to those with higher education. Other significant factors include residence, age of 1st marriage, religion, ethnicity, use of contraceptives
Socio-cultural factors influencing adolescent pregnancy in rural Nepal	Shrestha S. et al. [40]	2002	Nepal	Retrospective **Fair**	575 adolescent mothers (aged under 19 years at first pregnancy). Comparing 575 mature mothers	Community and educational empowerment	Socio-economic factors are presented in relation to their influence on pregnancy.	Comparatively adolescents had parents/elders decide majority of adolescent marriages
Perceptions of contraception, non-protection and induced abortion among a sample of urban Swedish adolescent girls: Focus group discussions	Thorsen et al. [41]	2006	Sweden	Qualitative (4 focus group discussions) **Fair**	16 urban adolescent girls aged 15–18 years.	Educational empowerment	Attitudes toward contraception, induced abortion and non-protection	Themes were: a) contraception: need for accessibility and affordability b) induced abortion: increased rate linked with younger sexual initiation, c) non-protection: alcohol use, partners reluctant to use condoms, common sex abuse, need for quality sex education in school.

^1^ Offers final quality assessment done by authors; ^2^ Analysis only conducted for adolescents with prior sexual intercourse with a male; ^3^ Reproductive Health Attitudes and Behavior (RHAB) Questionnaire used to assess both intervention (IG) and control (SC) groups every 3 months from baseline to 9 months; ^4^ SOC—Standard of Care; ^5^ RR—Rate Ration, CI—Confidence Interval; ^6^ STI/HIV—Sexually Transmitted Diseases/Human Immunodeficiency Syndrome.

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
