# Peer review of "Girls’ Empowerment and Adolescent Pregnancy: A Systematic Review"

_ijerph, 2020, doi:10.3390/ijerph17051664_

Round 1

Reviewer 1 Report

I appreciate the authors’ efforts to strengthen this manuscript and respond to reviewer feedback. However, the manuscript remains poorly organized and written and does not justify the limited scope of the review. More importantly, based on a closer look at the articles selected for this review, the reviewed studies do not consistently assess empowerment programs, which is the stated goal of this review. Instead, some of these studies assess associations between contexts such as poverty or teens’ perceptions of barriers to empowerment or abortion. This concern is a critical one which calls into question the purpose of the study and how it is operationalized and described.

Below are some more detailed comments and concerns.

Several paragraphs are not clearly written and it is difficult to understand the key points the authors are trying to express (e.g., lines 75-83).

There are many errors in language even in the first few paragraphs. For example, “lee” instead of “less” in Line 23 in the abstract, and language that doesn’t make sense in line 54. There also continue to be over-statements in the introduction and discussion. For example, both lines 52 and lines 56-57 involve over-statements, suggesting that most adolescent mothers have these struggles rather than some.

Line 59 about sustainable development needs explanation.

While the authors’ rationale for their focus on empowerment programs is improved, they still need to provide a clear explanation that addresses how empowerment may help to reduce teen pregnancy. For example, “this review focuses on adolescent empowerment programs because…” This can be placed after line 64.

The reviewer feedback on the need for connection between girl empowerment programs and sex educations programs (line 65) has not been clearly addressed. In this same paragraph, the focus on economic and sex education programs needs to tie directly to empowerment. Are these programs explicitly empowerment programs or other types of programs which can empower girls? The definition for an empowerment program needs to be clear and plays a key role in determining which programs are included in this review.

The authors have strengthened their case for how this study contributes to existing research.

In the methods section, the eligibility criteria for study inclusion (starting on line 113) do not include a program focus on empowerment. It is unclear how the Search Strategy plan was developed and how it relates to empowerment programs.

The authors did a good job explaining their quality assessment in the methods section and adding it to the results.

The discussion continues to over-state findings for the effects of empowerment, suggesting causal effects of empowerment when many reviewed studies are cross-sectional or qualitative.

Author Response

We appreciate your time and effort. Attached are the specific answers to the comments provided earlier.

Reviewer 2 Report

This thesis is only gut introduction to empirical research. I encourage you to do your own empirical research

Author Response

We appreciate your comments. Attached is the answer to the comment made earlier.

Round 2

Reviewer 1 Report

I appreciate the authors responses to my feedback.

However, they do not address the most critical feedback, which is:

"Further, based on the lack of clarity of the review paper selection, I looked more in depth at the articles the authors selected for review. From this exploration, it appears that the articles the authors selected for review do not consistently assess empowerment programs, in contrast with the stated purpose of the manuscript."

Several of the articles included in the review do not include programs, which is goes against the stated purpose of the review.

Author Response

Articles in the review, indeed do not report empowerment programs of which the objective clearly states. In retrospect, the original idea behind the review was to focus on empowerment and it affect pregnancy rates of adolescent girls. With this in mind, we have several changes to the introduction, purpose and discussion sections. More details are in the file attached

This manuscript is a resubmission of an earlier submission. The following is a list of the peer review reports and author responses from that submission.

Round 1

Reviewer 1 Report

This paper addressed an important issue of how interventions related to girls’ empowerment may impact teen pregnancy. However, its impact is limited by the small number of papers reviewed (9). Further, this paper is poorly organized, and needs a stronger rationale and significant English language editing.  Given the small sample size, the findings are also over-stated.

Given that there are many types of programs designed to reduce teen pregnancy, it would be helpful to explain why this paper focuses on empowerment programs specifically.

The second paragraph (lines 45-50) doesn’t read well. It jumps from point to point without a clear focus.

Paragraph 3 (lines 51-62) mixes findings from empowerment programs and sex educations programs, which are not the same.

Paragraph 4 (lines 63-70) focuses on reviews of adult women’s empowerment programs and does not include pregnancy outcomes. The authors need to clarify how women’s empowerment compares to girls’ empowerment and why these studies are relevant given they do not focus on pregnancy outcomes.

The authors need to make a stronger case of how this review fits with existing research and how it contributes to our understanding of girls’ empowerment and teen pregnancy.

The authors should clarify how they identified the four levels of girls’ empowerment assessed in their review: community, education, policy, support, economic.

It is helpful to include quality study measurements (lines 126-136). However, it would strengthen the paper to include this assessment in the results section (possibly in the table) rather than only providing information about study quality in the discussion.

Table 1 shares useful information, but the Major Findings section for each paper needs to be more clearly and concisely explained, with a focus on study findings only, not recommendations for intervention.

The discussion states, “Most studies showed that empowerment exerted its effect on adolescent sexual behavior which later reduced rates of adolescent pregnancy” (line 189-90) but this is not clear from the results section. Results should clearly demonstrate the patterns raised in the discussion.

The authors’ description that all 9 studies showed positive effects of empowerment on teen pregnancy (line 176) overstates study findings, particularly given several reviewed studies are qualitative. Statements comparing different types of empowerment are also over-stated, given the small sample size (line 177-178).

The breakdown of discussion sections into the four types of empowerment seems overly detailed, as only educational empowerment has a reasonable number of studies to provide initial conclusions about the area, while other components are only includes in 2-3 studies. 

The authors recognize the limitation of including only 9 studies in the review. They write that this may “reflect the scope of the search strategy as only three databases were used…” (lines 287-288). This raises the question of why the authors limited their review to three databases and whether they could broaden the scope of their research to produce a more comprehensive review.

Reviewer 2 Report

Topic is of great interest.

English language is adequate.

Introduction is well organized.

Material and Methods are well structured.

Results are well presented.

Discussion is fine demonstrated. The authors must state what do they think this study will change in the near future in favor of avoidance of unintended adolescent pregnancies and how situation in intended adolescent pregnancies will be improved by this study.

Reviewer 3 Report

The work presents a Systematic Review of the articles on the topic: Girls' Empowerment and Adolescent Pregnancy.

the work is clearly exempted.

with regard to the considerations, the conclusions in relation to the various components of empowerment identified in the articles should be implemented.the different levels of empowerment could be considered to propose an intervention in an ecological perspective (see articles: Di Napoli, I., Procentese, F., & Arcidiacono C., (2019) Women's associations and well-being of their members: from mutual support to full citizenship. Journal of Prevention and Intervention in the Community, 14 (2). https://doi.org/10.1080/10852352.2019.1624357.Di Napoli, I., Procentese, F., Carnevale, S., Esposito, C., & Arcidiacono, C. (2019). Ending intimate partner violence (IPV) and locating men at stake: an ecological approach. International journal of environmental research and public health, 16 (9), 1652

Reviewer 4 Report

The is no clear research goal.

The article is a synthesis of literature. As such, it can be a good introduction to research project.It lacks critical analisys of the text.

The conlusions are too general. They should be considered as general descriptions, because they are only the summary of previous studies.

The researches treated the results of quantitative and qualitative research as equivalent, wchich should be considered a methodological mistake.